**Observations of Aerosol-Vapor Pressure Deficit-Evaporative Fraction coupling over India**
Chandan Sarangi[1,2,7*], Tirthankar Chakraborty[3,4], Sachchidanand Tripathi[1,3*], Mithun Krishnan[1],
Ross Morrison[5], Jonathan Evans[5], Lina Mercado [5,6]
Affiliations:
[1] Department of Civil engineering, Indian Institute of Technology, Kanpur, Kanpur, India
[2] Department of Civil engineering, Indian Institute of Technology, Madras, Chennai, India
[3] Center for Environmental Science and Engineering, Indian Institute of Technology, Kanpur
[4] School of the Environment, Yale University
[5] UK Centre for Ecology & Hydrology, Wallingford, UK
[6] Department of Geography, University of Exeter, UK
[7] Laboratory of Atmospheric and Climate Sciences, Indian Institute of Technology, Madras,
Chennai, India
* Corresponding authors: snt@iitk.ac.in and chandansarangi@iitm.ac.in
**Abstract**
North India is a densely populated subtropical region with heavy aerosol loading (mean Aerosol
Optical Depth or AOD ~ 0.7), frequent heatwaves and strong atmosphere-biosphere coupling,
making it ideal for studying the impacts of aerosols and temperature variation on latent heat flux
(LH) and evaporative fraction (EF). Here, using in situ observations during the onset of the
summer monsoon over a semi-natural grassland site in this region, we confirm that strong co-
variability exists among aerosols, LH, air temperature ($T_{air}$) and vapor pressure deficit (VPD).
Since the surface evapotranspiration is strongly controlled by both physical (available energy and
moisture demand) and physiological (canopy and aerodynamic resistance) factors, we separately
analyze our data for different combinations of aerosols and $T_{air}$/VPD changes. We find that
aerosol loading and warmer conditions both reduces SH. Further, we find that an increase in
atmospheric VPD, tends to decrease the gross primary production (GPP) and thus LH, most
likely as a response to stomatal closure of the dominant grasses at this location. In contrast, under
heavy aerosol loading, LH is enhanced partly due to the physiological control exerted by the
diffuse radiation fertilization effect (thus increasing EF). Moreover, LH and EF increases with
aerosol loading even under heatwave conditions, indicating a decoupling of plant's response to
VPD enhancement (stomatal closure) in presence of high aerosol conditions. Our results
encourage detailed in situ experiments and mechanistic modelling of AOD-VPD-EF coupling for
better understanding of Indian monsoon dynamics and crop vulnerability in a heat stressed and
heavily polluted future India.

**Highlights:**
1. A rigorous analysis of Aerosol-EF-VPD coupling using collocated direct observations is
presented
2. Increased aerosol loading enhances Evaporative Fraction by decreasing sensible heat and
increasing latent heat.
3. Aerosols modulate the response of vegetation to changes in VPD under heatwave conditions

**Keywords:** Grassland, Aerosol loading, eddy covariance, evaporative fraction, physiological
response, diffuse radiation, Indo Gangetic Plains, heatwave, sensible heat, latent heat, Bowen
ratio
**Introduction:**
The surface energy balance represents the balance between the net radiation (NR) flux at
the Earth's surface and the partitioning of NR into latent heat (LH), sensible heat (SH) and
ground heat (GH) fluxes [Wang and Dickinson, 2012]**.** While the dominant partitioning of
energy as SH enhances the near-surface air temperature, the LH flux cools the surface and
increases the moisture content of the boundary layer. Thus, perturbations to the partitioning of
the outgoing turbulent energy fluxes from the land surface modify the near surface
micrometeorology. One way of representing this partitioning is the evaporative fraction
(EF=LH/(SH+LH)), or the proportion of the total available energy (NR-GH) available at the
surface released via vegetation evapotranspiration and soil evaporation. Earlier studies have
established that the EF can be modulated by a range of factors, including vapor pressure deficit
(VPD), soil moisture, canopy structure, atmospheric composition, solar radiation and stomatal
behaviour [Baldocchi, 1997; Wilson et al., 2002].

The variability in VPD, which describes the near surface moisture deficit for a given

temperature (difference between the saturated and ambient vapor pressure for atmospheric water)
is arguably the dominant nonlinear forcing on EF variability [Gu et al., 2006]. On one hand, an
increase in VPD leads to the partitioning of more of the available energy into LH to meet the
atmospheric moisture demand, part of the physical control on evapotranspiration [Penman, 1948;
Monteith et al., 1965]. On the other hand, high VPD also triggers partial closure of leaf stomata
in response to increased atmospheric dryness [Jones and Sutherland, 1991; Damour et al., 2010;
Medlyn et al., 2011]. This is part of the physiological control on ET, causing an increase in VPD
to actually decrease ET (and thus EF) [Rigden & Salvucci, 2017].  Moreover, the sign of VPD-
EF association could also change due to variations in confounding factors like ambient soil
moisture and diffuse/direct radiation [Gu et al., 2006]. More diffused radiation enhances plant
productivity [Mercado et al., 2009; Rap et al., 2018] and plant growth [Wang et al., 2018];
which, in turn, can increase LH and EF [Chakraborty et al., 2021;Davin et al., 2012; Wang et al.,
2008]. However, this association is also reported to have an optimum point beyond which plant
productivity declines with increasing diffused fraction of radiation [Knohl et al., 2008].

Small particles suspended in the atmosphere, i.e. atmospheric aerosols, can alter the

amount of shortwave and longwave radiation reaching the surface, through scattering and
absorption, thereby altering NR [Schwartz, 1996; Trenberth et al., 2009; Chakraborty and Lee,
2019]. This is commonly known as the aerosol direct radiative effect (ADRE) and is dependent
on aerosol size, composition and vertical distribution in the atmosphere [Forster et al., 2007;
Sarangi et al., 2016]. Global and regional scale modelling studies have reported that the ADRE
can greatly alter the surface fluxes and microclimate over land [Liu et al., 2014; Mallet et al.,
2009; Shen et al., 2020; Myhre et al., 2018]. Generally, the ADRE reduces NR, which results in
the reduction in the magnitude of SH and LH. But, loading of scattering aerosols from fossil fuel
combustion can also increases the diffuse fraction of solar radiation at the surface, which affects
the photosynthesis and LH or EF [Chameides et al., 1999; Matsui et al., 2008;Niyogi et al., 2004;
Wang et al., 2008; O'Sullivan et al., 2016; Wang et al., 2020]. This mechanism is generally
referred to as the diffuse radiation induced aerosol fertilization effect (ADFE). But, depending on
the ecosystem, the positive association of ADFE on EF also gets saturated as ADRE becomes
larger than a threshold [Yue et al., 2017]. Further, Steiner et al., [2013] reported that warmer air
temperature are consistent with high aerosol optical depth (AOD) scenario over various in-situ
micrometeorological sites in USA, which can result in no clear association between AOD and
LH. Thus, how aerosol loading modulates the already complex VPD-EF association can depend
on the interplay between radiation, ADFE, aerosol amount and properties, background climate
and ecosystem phenology [Steiner et al., 2011].
Northern India is a global hot spot for atmospheric aerosols with AOD varying between 0.5 and
1.5, and high aerosol radiative efficiency values ( ~100 W/m$^2$/AOD) during pre-monsoon period
[Dey et al., 2011; Kumar et al., 2015; Dimitris et al., 2012; Sarangi et al., 2016; Srivastava et al.,
2011]. In addition, the region also experiences frequent high temperature days and heatwave
conditions, generally extending for 2-6 days during this period [Ratnam et al., 2016; Rohini et
al., 2016]. During heatwave conditions, the regional atmosphere is largely stagnant [Ratnam et
al., 2016], which can lead to greater air temperature by 5-10 K and magnifies the water vapour
demand by 2-3 times at weekly time scale. In addition to high air temperatures ($T_{air}$), high
aerosol loading during heatwaves have also been reported over Northern India [Dave et al., 2020;
Mondal et al., 2020] at this time of year. Moreover, the value of EF is typically greater than 0.5
over the Northern India during pre-monsoon period, indicating a potentially larger control of
VPD-LH linkages on surface energy partitioning [Bhat et al., 2019]. Steep variability in ambient
values of VPD (also AOD in some events) during heatwaves over Northern India provides us
with ideal conditions for investigating the associations between aerosol loading and VPD-EF
coupling.

Previous studies have suggested that aerosol loading can modulate the partitioning of surface
fluxes over Northern India [Urankar et al., 2012; Murthy et al., 2014; Latha et al., 2019; Gupta et
al., 2020]. However, these studies have been based on reanalysis products [Urankar et al 2012],
very limited measurements of SH only [Murthy et al., 2014] or estimated derived from remotely
sensed data [Latha et al., 2019] and therefore lack the fidelity that can be obtained from direct
observations of key processes. Better understanding of the aerosol-VPD-EF associations using
direct collocated observations is essential to understand present day conditions and potential
feedbacks that can modify future climate over this region of great hydro-climatic significance. In
this study, we have used co-located observations of surface energy balance, near-surface
micrometeorological variables and soil characteristics, together with aerosol properties (both
surface and columnar) at a sub-tropical site in northern India during the pre-monsoon season.
Analysis of case studies with AOD varying in phase or remaining constant with high VPD
(under heatwave conditions) are done to understand the underlying processes. Here, we will
present compelling evidence that changes in EF is directly (indirectly) proportional to aerosol
loading (VPD). More interestingly, we found that aerosol loading can decouple the observed
strong VPD-LH relationship under heatwave scenario which can have serious implications on
climate resilience of crops and vegetation. Below, the sections are organized to discuss the data
used, case studies selected and methodology, results, discussions and summary of this study.
**2. Observation site and data:**
Observations of SH, LH and net ecosystem $CO_2$ exchange (NEE) were obtained over a
semi-natural grassland site (Figure 1A) within the campus of the Indian Institute of Technology,
Kanpur (IITK; 26.5N, 80.3E, elevation 132 m above mean sea level) during the pre-monsoon
months (April-June) of 2016-2017. Energy flux data were collected by an eddy covariance
system installed at 5.28 m above the soil surface. This flux measurement site is part of an eddy
covariance network set up in India as part of the INCOMPASS project of the Indo-UK Monsoon
Programme [Chakraborty et al., 2019; Turner et al. 2019; Bhat et al., 2019]. The eddy covariance
system consists of a Windmaster sonic anemometer-thermometer (Gill Instruments Ltd.
Lymington, UK) and a LI7500 infrared gas analyzer (LI-COR Biosciences, Logan, Utah, USA).
The fetch around the tower is a mixture of different C4 grasses, i.e. variants of Napier grass
(~60-70%) and some common reed (Scientific family: Pennisetum purpureum and Phragmites-
Saccharum-Imperata). Napier grasses are invasive and a perennial species and representative of
grasslands in the region (Chakraborty et al., 2019; Holm et al., 1979). The vegetation cover is
more than 90% of the fetch of the flux tower (Figure 1B) and the canopy height varied within 1-
1.5 m during our study periods. The soil is typical of the Gangetic Plains with silt, clay and sand
fractions of 80%, 15% and 5%, respectively (unpublished data). The site experiences a humid
subtropical climate. The range in daily AOD and $T_{air}$ was 0.4-1.4 and 32-45 $^{O}$C, respectively,
during the study period (Figure 1C).

The net radiation (NR; W m$^{-2}$) and its incoming and outgoing short- and longwave

components were measured using an NR01 net radiometer (Hukesflux, Delft, The Netherlands)
installed at 5 m above the surface. The surface temperature ($T_{srf}$) was calculated from the
measured outgoing longwave radiation following the Stefan–Boltzmann law assuming an
emissivity of 0.95 [Trenberth et al., 2009]. Ground heat fluxes (GH; W m$^{-2}$) were  monitored at
0.03 m below the soil surface using two HFP01-SC self-calibrating soil heat flux plates
(Hukesflux, Delft, The Netherlands). Near surface air temperature ($T_{air}$; $^{o}$C) and relative humidity
(RH; %) were measured at a height of 4.5 m. Wind speed and wind direction were  measured at
10 m above the soil surface using a WindSonic anemometer (Gill Instruments Ltd., Lymington,
UK). Volumetric soil water content (VWC; m$^3$ of water in m$^3$ of soil) and surface temperature
($T_{srf}$; $^{o}$C) were measured using two pairs of digital TDT sensors (Acclima Inc., Meridian, Idaho,
USA) installed at 0.05 and 0.15 m below the soil surface. Standard data processing and quality
control routines were used to calculate surface fluxes as described in Morrison et al. 2019. Data
gap-filling and the partitioning of net ecosystem exchange into Gross Primary Production (GPP)
and total ecosystem respiration was performed using the R EddyProc package [Reichstein et al.,
2016; Reichstein et al., 2005]. Negative net ecosystem exchange during the daytime period
indicates that photosynthesis at our site dominates over soil and plant respiration (not shown).
Since water and carbon cycles in the plants are closely coupled [Collatz et al., 1991]; variations
in GPP are used as a proxy for plant transpiration in this study. More details on the flux, weather
and radiation tower measurements at IIT Kanpur can be found in Table S1 and Chakraborty et
al., 2019.

Version 2 instantaneous cloud screened (Level 1.5) half-hourly averages of Aerosol

Optical Depth (AOD) at 550 nm and Single Scattering Albedo (SSA), the ratio of scattering
efficiency to total extinction efficiency, at 440 nm obtained from the AErosol RObotic NETwork
(AERONET) station deployed in the IITK campus (Figure 1A) were used to quantify the aerosol
optical properties during our study period. Low and high SSA values indicate dominance of
absorbing and scattering aerosols in the column, respectively. Clear-sky short wave (0.25–4μm)
radiative transfer calculations, using the Santa Barbara discrete ordinates radiative transfer
Atmospheric Radiative Transfer Model (SBDART) [Ricchiazzi et al., 1998], are used to estimate
the midday aerosol direct radiative forcing (ADRF) at surface and diffuse radiation reaching the
surface (diffuse$_{frac}$).  Midday mean AOD and SSA for each day are prescribed to the model.
More details on radiative flux calculations using SBDART are mentioned in Supplementary
Information file. Finally, micro-pulse lidar backscatter images (Level 1.5) measured at the
collocated Micro-Pulse Lidar Network site [Campbell et al., 2002; Welton and Campbell, 2002]
are also used in this study, mainly to identify cloudy days. A day is termed as a cloudy day if
cloud patches are observed in Lidar profiles for more than 3 hours. More details on the aerosol
measurements can be found in supplementary information file.

**3. Case studies and methodology:**

In order to examine the impact of aerosols or VPD on EF, we need to carefully identify

periods where the variability of other confounding factors is negligible. As such, we identified
three weeks (marked in Figure 1C) for analysis, where daily variations in all these factors except
$T_{air}$ /VPD and AOD is negligible. Figure 1C illustrates the occurrences of cloudy days, rainfall
and wildfire-affected periods during pre-monsoon months of 2016 and 2017. We have avoided
periods of cloud and rainfall occurrences since that would affect the surface and energy budget
much more than the ADFE.  The daily mean VWC values are also shown for the period in Figure
1C.  However, as shown in Figure 1C, it is rare to have a considerable time interval with only
variation in AOD values (and negligible variation in $T_{air}$/VPD). Eventually, three one-week
periods are carefully selected with different combinations of dominant weekly gradients in $T_{air}$
/VPD and AOD and analyzed to gain insights into ambient AOD-VPD-EF association. The first
week selected for analysis is between 2$^{nd}$-9$^{th}$ June, 2016, which had high weekly gradient in
AOD but was accompanied by low variation in $T_{air}$/VPD (hereafter referred as High AOD-Low
$T_{air}$ (HALT) case). The second week is during 10$^{th}$-15$^{th}$ April, 2017, which witnessed large daily
increase in aerosol loading as well as $T_{air}$ in phase throughout the week (hereafter referred to as
the High AOD-High $T_{air}$ (HAHT) case). We also selected a third week during 10$^{th}$-15$^{th}$ May,
2017, when high gradient in $T_{air}$ was observed across the week, but negligible weekly gradient in
AOD was present i.e the AOD values had large day to day variability through the week
(hereafter referred to as the Low AOD- High $T_{air}$ (LAHT) case). Interestingly, heatwave
conditions were prevalent over North India during the HAHT and LAHT weeks, therefore, a
wide range of VPD-AOD-EF variation can be sampled. Moreover, since there were no rainfall
events during these three weeks, the variation in VWC was minor compared to large daily
variations in $T_{air}$ and AOD during our study periods. Further, the variations in the vegetation
phenology, wind and boundary layer height are found to be negligible within each of these three
weeks. Note that no week with low AOD and low VPD variations was observed during our study
period.
The simultaneous midday (1000-1500 LT) variability in AOD, VPD, EF and the other
components of the surface radiative balance is analyzed across the HALT and LAHT weeks to
understand the impact of strong weekly gradients of AOD and VPD, respectively. Further, we
analyse the weekly gradients in the observations during HAHT, and compare and contrast the
same with the HALT and LAHT cases to understand the combined effects of AOD and VPD.
Moreover, to examine the impact of aerosol loading on VPD-EF associations under enhanced
heat stress, we also calculated the daily midday bulk canopy resistances for both HAHT and
LAHT cases by inverting the Penmann-Monteith equation as described below. We used observed
values of available energy, VPD, $T_{srf}$ derived from observed $LW_{out}$, psychrometric constant and
slope of vapor pressure curve derived from observed surface pressure and $T_{air}$ respectively, and
aerodynamic resistance derived from the observed SH and near-surface temperature gradient.
The aerodynamic resistance to heat transfer ($r_a$) is calculated from the near-surface temperature
gradient and the measured distance between the two (H), given by:
$$r_a = \frac{-\rho C_p\ (T_{srf} - T_{air})}{H}$$
where $T_{srf}$ is the surface temperature, calculated by inverting the Stefan-Boltzmann law assuming
a unit surface emissivity (reasonable for vegetated surfaces), $\rho$ is the air density, and $C_p$ is the
specific heat at constant pressure ($1.005 \times 10^{-3}$ MJ kg$^{-1}$ °C$^{-1}$).
Then, the canopy resistance ($r_s$) is calculated by inverting the Penman-Monteith approximation.
Thus:
$$r_s = \frac{\left(\dfrac{\Delta(Rn-G)+\dfrac{\rho CpVPD}{r_a}}{LE}\right)-\Delta}{\gamma-1}r_a$$

where $\Delta$ is the slope of the water vapor saturation curve given by:
$$\Delta = \frac{4098\left[0.6108\exp\left(\frac{17.27T_a}{T_a+237.3}\right)\right]}{(T_a+237.3)^2}$$

and $\gamma$ is the psychrometric constant, calculated as:
$$\gamma = \frac{C_p P}{\varepsilon\lambda}$$

where P is atmospheric pressure in kPa, $\lambda$ is the latent heat of vaporization (2.45 MJ kg$^{-1}$), and $\varepsilon$
is the ratio of the molecular weight of water vapour to dry air (0.622).
**4. Results:**
During the HALT period, midday AOD values decreased monotonically across the week from
~1.1 on 2$^{nd}$ June, 2016 to ~ 0.6 on 9$^{th}$ June,2 016 (Figure 2A). The corresponding trend in SSA
values was negligible, but SSA values are ~0.92 indicating a predominance of scattering aerosols
(Figure 2A). Corresponding values of NR at surface increased monotonically by ~50 W/m$^2$
during the same week (Figure 2D). The enhancement in midday NR with decreasing AOD is
strongly driven by the corresponding increase in midday incoming shortwave radiation (ISWR)
by ~100 W/m$^2$ (Figure 2D). In agreement, ADRF values at surface decreased by ~80 W/m$^2$ and
diffuse fraction of incoming radiation increased by ~0.10 with decrease in scattering aerosols
from 2$^{nd}$ June to 9$^{th}$ June, 2016 (Figures S1A and S1D).  The daily trend in modelled ADRF (and
diffused fraction) values are consistent with the daily reduction trend of ISWR during HALT,
reinforcing the expectation that negative daily trend in ISWR and NR during HALT was
primarily by aerosol-induced radiative changes.

During HAHT, the midday AOD values increased monotonically across the week from

~0.3 on $10^{th}$-$11^{th}$ April to ~ 0.8 on $14^{th}$-$15^{th}$ April (Figure 2B). Corresponding values of NR and
ISWR at surface decreased monotonically by ~100 $W/m^2$ and ~200 $W/m^2$, respectively, during
the same period (Figure 2E). Similar to HALT, no daily trend was present in SSA values during
HAHT and SSA values are ~ 0.9 indicating presence of scattering aerosols (Figure 2B). In
agreement, ADRF values at surface decreased across the week (Figure S1B) with highest values
on high AOD days ($14^{th}$-$15^{th}$ April; ~150 $W/m^2$) compared to those on low AOD ($10^{th}$-$11^{th}$ April;
~50 $W/m^2$). At the same time, the diffuse fraction of incoming radiation at the surface (Figures
S1E) increased substantially from ~ 0.5 (on $10^{th}$ April) to ~0.7 on ($15^{th}$ April) during HAHT
indicating strong impact of aerosol loading.

In contrast, during LAHT week, the gradient of AOD values from $10^{th}$ and $15^{th}$ May, 2017 was
relatively minor (Figure 2C). As the increase in AOD through the week was smaller compared to
other two cases, corresponding decrease of NR and ISWR values at surface was also smaller in
magnitude (~30 $W/m^2$) during this period (Figure 2F). Correspondingly, negligible trend in
ADRF (Figures S1C) at the surface is observed indicating low variation in aerosol radiative
effect change during the LAHT week. Moreover, the midday SSA values during LAHT are
lower (~0.8) compared to HALT and HAHT cases indicating presence of highly absorbing
aerosols in the column (Figure 2C). Accordingly, the ADRF values at surface during LAHT
(Figure S1C) were very high, more than double of the same during HALT and HAHT (i.e. ~350
$W/m^2$). This can be explained by the fact that absorbing aerosols (lower SSA values) were
relatively dominant during LAHT compared to the other 2 cases. Moreover, dominance of
absorbing aerosols also lead to minor variation in diffused radiation during the week (Figure
S1F). To sum up, the impact of aerosol variability (i.e. the gradient in direct radiative effect and
diffused fraction modulation) is minor during the week compared to HAHT and HALT weeks.

As aerosol direct radiative effect induces surface cooling, midday $T_{srf}$ values reduced

from ~ 35°C during low AOD days to ~30°C during high AOD days across the HALT week
(Figure 3A). At the same time, the variability in $T_{air}$ values remain more or less constant during
HALT. Therefore, the midday variation of temperature difference between $T_{srf}$ and $T_{air}$ ($\Delta T = T_{srf}$
- $T_{air}$) is inversely proportionally with aerosol loading for HALT (Figure 3A). Greater the value
of $\Delta T$, greater will be the turbulent and convection flux, and greater is the tendency of SH flux
release at surface. Consequently, sensible heat fluxes are also inversely proportional to increase
in AOD (and aerosol direct effect). With increase (decrease) in $\Delta T$ (AOD) values, the
corresponding SH values increased linearly from ~60 W/m$^2$ on 2$^{nd}$ June to ~ 120 W/m$^2$ on 9$^{th}$
June, 2016 during HALT week (Figure 3D).

By contrast, a distinct and steep increase in midday $T_{air}$ (~10 $^o$C) is seen during HAHT

and LAHT weeks. Correspondingly, the mid-day $T_{srf}$ values are also seen to be increasing in
close coupling with the $T_{air}$ values during these two weeks (Figures 3B-C). This coupling is
mainly because of the coexisting stagnant scenario under heatwave periods. Nonetheless, $\Delta T$
variation is inversely proportional to AOD variation during both the weeks (Figure 3B-C).
Because, some portion of the enhancement in midday $T_{srf}$ is compensated by the aerosol-induced
surface cooling, steeper AOD trend across the week means greater $\Delta T$ magnitude. For instance,
as aerosol radiative effect is relatively smaller across the week during LAHT compared to that
during HAHT, a relatively larger decrease in daily $\Delta T$  (> 2 $^o$C) is observed during HAHT week
(Figure 3B). Consistently, the magnitude of SH also significantly decreased across the week in
HAHT and LAHT. Specifically, the midday mean values of SH decreased linearly from ~200
W/m$^2$ on 10$^{th}$ April (low AOD) to ~ 100 W/m$^2$ on 15$^{th}$ April, 2017 (high AOD) during HAHT
(Figure 3E). During LAHT, the midday mean SH decreased linearly from ~200 W/m$^2$ on 11$^{th}$
May to ~ 125 W/m$^2$ on 14-15$^{th}$ May, 2017 (Figure 3F).

The midday latent heat values decreases by ~150 Wm$^{-2}$ from high AOD days to low

AOD days during HALT week (Figure 3D). In comparison, the increase in LH values with
increase in AOD across the HAHT week from 10$^{th}$ April,2017 to 15$^{th}$ April, 2017 is gradual i.e.
~25 W/m$^{-2}$ (Figure 3E). Specifically, the slope of regression of latent heat against AOD is 70
W/m$^2$/AOD and 10 W/m$^2$/AOD for HALT and HAHT cases, respectively (figure not shown).
As, VPD values increase steeply in HAHT case (Figure 3H), but no distinct variation in VPD
across the week was evident for HALT case (Figure 3G). Examination of corresponding midday
values of gross primary production (GPP) flux (Figures 3G-F) also illustrate gradients similar in
sign to corresponding latent heat fluxes indicating that the daily variation in LH flux in both the
cases is mainly due to associated variation in evapotranspiration. Keeping in mind that the
magnitude of AOD variation in both the above cases are similar, the differences in slopes of LH-
AOD regression (lower value during HAHT) could be attributed to the simultaneous suppression
of evapotranspiration by VPD rise during HAHT week.

VPD-associated decline in GPP and thus LH fluxes is even more clearly observed during

LAHT week. A strong negative trend in midday values of latent heat and GPP is observed as the
week progressed from low to high VPD during LAHT (Figure 3F and 3I). Quantitatively, the
slope of regression of (midday mean) latent heat against $T_{air}$ is +4.1 W/m$^2$/$^o$C and -6.6 W/m$^2$/ $^o$C
for HAHT and LAHT cases, respectively. Note that the magnitude of VPD variation in both the
cases is similar, so the differences in slope of latent heat and $T_{air}$ regression can be attributed to
the corresponding differences in aerosol loading. Thus, the magnitude of latent heat or GPP is
directly proportional to changes in magnitude of AOD (as seen in HALT), but the same is
inversely proportional to variations in $T_{air}$ or VPD (as seen in LAHT), and the net effects can
largely compensate each other (as seen in HAHT).

Moreover, the gradient in EF was substantial only in HAHT and HALT where there was
substantial variation in AOD across the week. Partitioning of surface energy into latent heat or
the latent heat fraction (LHF: Latent heat / Net radiation) decreased and that into sensible heat
fraction (SHF: Sensible heat / Net Radiation) increased with increase in AOD across the week
during HALT (Figure 3J). As a result, the midday EF distribution decreased with reduction in
AOD from ~0.8 on 2$^{nd}$ June to ~0.6 on 9$^{th}$ June during HALT (Figure 3J). On the same line,
with increase in AOD across the week during HAHT, EF also increased from ~0.63 on 10$^{th}$
April, 2017 to ~0.78 on 15$^{th}$ April, 2017 (Figure 3K) due to simultaneous decrease and increase
in SHF and LHF, respectively. But, in absence of clear aerosol gradient across the week, no
substantial variation was observed in EF across the week during LAHT case (Figure 3L). The
decrease in sensible heat with VPD enhancement was similar in HAHT and LAHT cases (Figure
3K-L). But, LH release increased (decreased) with VPD during the former (later) case indicating
a role of AOD change on VPD-EF association.

Figure 4 illustrates the variation in midday mean canopy resistance during the LAHT and
HAHT weeks to various physical and physiological factors that control evapotranspiration,
namely moisture demand, available energy, air temperature and the aerodynamic resistance. As
expected, the canopy resistance is significantly ($p<0.05$) correlated with VPD although clear
differences in the slope is present for the two cases. Specifically, the canopy resistance increases
steeply from 400 to 1400 s m$^{-1}$ with increase in VPD from 40 to 70 hPa during LAHT case
(Figure 4a). However, the canopy resistance only increases from 400 to 500 with an increase in
VPD from 45 to 65 hPa during HAHT case (Figure 4a). Similarly, air temperature during these
periods also shows a statistically significant positive relationship with canopy resistance (Figure
4d). However, during both periods, canopy resistance was found to be independent of available
energy (Figure 4c) and the aerodynamic resistance (Figure 4d), indicating that the sensitivity of
canopy resistance to changes in VPD (or $T_{air}$ ) is significantly greater than that for the other
variables.
The LAHT case illustrates the frequently reported behaviour of reduction of canopy
conductance under increasing VPD due to partial stomata closure as a physiological stress
response (Grossiord et al., 2020). Similar responses are also reported in Napier grasses, the
native vegetation over our site (Mwendia et al. 2016). Napier grasses can be anisohydric, i.e.
water spending under ample water availability (Cardoso et al., 2015). But their behaviour
becomes isohydric under high temperature and high water stress (Liang et al., 2017; Mwendia et
al. 2014; Purbajanti et al., 2012). During both HAHT and LAHT weeks,  soil moisture is very
low, hence, the Napier grasses behaves isohydrically under high VPD. The comparison of LAHT
and HAHT scatter illustrates that canopy conductance is not strongly affected even under severe
VPD rise when aerosol loading also increases in phase. Specifically, the strong gradient of
increase in canopy resistance with VPD/ air temperature gets moderated under the high aerosol
scenario. Thus, under the presence of high aerosol loading, the isohydric response of Napier
grass to temperature rise or the physiological stress under high VPD is decoupled. This can
partially explain the aerosol-induced increase in EF (as well as LH and GPP) even under high
VPD rise during HAHT.
Further, meteorological co-variability or any significant differences in weekly pattern of
other micro-meteorological variables between HAHT and LAHT cases can also contribute to the
corresponding differences in AOD-VPD-EF association. A closer look illustrates that minor
gradients are present in the meteorological variables (Figure S2), which can have secondary
effects on the VPD-EF associations. Nonetheless, the individual or relative contribution of these
meteorological variability and aerosols on the observed coupling remains unknown and deserves
further attention in future studies with in depth mechanistic modelling.

**5. Discussion:**

The increase in scattering aerosols increased diffused radiation during HALT; thereby
facilitating relatively more photosynthesis and thus more GPP and latent heat release with
increase in AOD. At the same time, increase in AOD also decreased the temperature difference
between surface and air and constrained sensible heat release, eventually leading to aerosol-
mediated increase in EF during HALT. However, previous studies investigating the role of
aerosols on surface energy fluxes over India have largely reported that aerosol loading is
inversely related to latent heat [Murthy et al., 2014; Latha et al., 2019; Gupta et al., 2020].
Possible explanations for this apparent contradiction are as follows. First, these studies did not
explicitly account for the effect of daily meteorology/ VPD/ temperature variability in their
analysis which can have confounding effects (as shown here and discussed in Steiner et al.,
2013). Second, these studies were not focused on grassland. Murthy et al., 2014 used
micrometeorological site data with a forested footprint in Ranchi. At the same time, Latha et al.,
2019 performs analysis at 100 km spatial resolution from reanalysis product/Model, which is
representative of a composite land use (including cities, forest, cropland and grassland) and thus
a mixture of evapotranspiration and ground evaporation. Gupta et al., 2020 used
micrometeorological observations within a typical university canopy (buildings, roads and trees)
in Mumbai. Note that total LH can decrease due to aerosols and EF can still increase if SH is
decreasing more than EF due to reduction in available energy. Nonetheless, our finding of direct
proportionality between aerosol loading and latent heat (or photosynthesis) is consistent with
previously reported in-situ studies over grasslands sites in USA [Niyogi et al., 2004; Gu et al.,
2002; Wang et al., 2008].
In contrast, aerosol loading and heatwave conditions both supressed sensible heat release.
Greater aerosol direct radiative effect induces more surface cooling (Chakraborty and Lee,
2019), and hence lower sensible heat fluxes (Yu et al., 2002; Urankar et al., 2012; Steiner et al.,
2013), as seen in HALT case. Simultaneously, sensible heat release is also directly proportional
to the near surface temperature gradient during Pre-monsoon (Rao et al., 2019), which is clearly
seen in LAHT case. In HAHT case, both the effects work in phase to supress release of sensible
heat. The reduction of sensible heat per unit change of $T_{air}$ is 8 W/m$^2$/$^o$C during LAHT
compared to the same being 11 W/m$^2$/$^o$C in HAHT case. At the same time, the reduction of
sensible heat per unit change of AOD is 135 W/m$^2$/AOD during LAHT compared to the same
being 65 W/m$^2$/AOD in HALT case. Hence, increase in AOD and $T_{air}$, both suppress the release
of available surface energy via sensible heat and the effect is largely additive. Moreover, the
intensity of the AOD-induced sensible heat suppression will be stronger if the aerosols are
composed of relatively more absorbing aerosols, specifically black carbon [Myhre et al., 2018].
Because, they not only cools the $T_{srf}$ (Mallet et al., 2009; Pandithurai et al., 2008a; Shen et al.,
2020) but also can warm $T_{air}$ (especially under stagnant/heatwave conditions), thereby reducing
the near surface temperature gradient and inducing lower tropospheric stability [Dave et al.,
2020; Steiner et al., 2013; Myhre et al., 2018].

However, contrary to our results, a recent modelling study over India reports that

enhancement of absorbing aerosols are positively associated with increase in sensible heat and
air temperature under heatwave scenario [Mondal et al., 2020]. The inherent model biases in the
aerosol properties and concentration as well as absence of detailed canopy-atmosphere processes
in the model simulations of Mondal et al., 2020 may cause differences in the signature of the
AOD-sensible heat feedback. At the same time, the above differences can also be explained by
taking into consideration the difference in time-scale of the feedback used in analysis. For
example, a robust positive association between morning time black carbon concentrations and
mid-day $T_{air}$ is observed by Talukdar et al., 2020. Although, they attributed this association
primarily to diurnal evolution of the residual layer mixing, the understanding from our study can
also explain a possible pathway. High black carbon loading during morning time can suppress
instantaneous sensible heat release (via reduction in the near surface temperature gradient),
followed by release of the additional sensible heat amount in the mid-day period under relatively
unstable atmosphere (and lower black carbon concentration due to dilution effect). As such,
correlations between absorbing aerosols and sensible heat at instantaneous scale can be negative
(as seen in HAHT), but correlations or composite analysis at daily or monthly time scale may
involve feedbacks which can result in positive associations (as also seen in Mondal et al., 2020).

In addition, our results clearly underline the complexity and non-linearity between

aerosol, VPD and EF, and provides observational evidence to the discussions reported in Steiner
et al., 2011; 2013. Keeping all other factors relatively constant, increase in scattering aerosols
causes a positive AOD-EF association (as seen in HALT). In case of HAHT, as both AOD and
VPD increased in phase over the week, VPD-induced reduction in evapotranspiration
compensated a major portion of the aerosol fertilization effect resulting in a slight increase in
latent heat with increase in AOD. Also note that, combined effect of increase in AOD and $T_{air}$
caused a large suppression in sensible heat fluxes. Thus, EF also increases with AOD under
heatwave conditions. However, in absence of significant aerosol variation, the increase in VPD
causes a large reduction in evapotranspiration (as seen in LAHT). First, negligible aerosol
fertilization effect and second, increase in canopy resistance (via stomatal aperture reduction)
under steep rise in VPD values caused large reduction in latent heat across the week during
LAHT. High VPD is also linked with greater $T_{air}$ during heatwave scenarios, thereby inducing
reduction the near surface temperature gradient and sensible heat during LAHT. Thus, both
sensible heat and latent heat release decreased with VPD causing negligible change in EF with
VPD. Thus, the VPD-EF coupling is very strong in absence of aerosol loading but weakens
under aerosol loading. Along with aerosol fertilization effect, the direct deposition of aerosols as
a wax layer on the leaf surface can also contribute to such an effect [Burkhardt., 2010; Burkhardt
and Grantz., 2017]. Recently, Grantz *et al*. 2018 used direct observations in glasshouses to
illustrate decoupling of stomata conductance (flux-based) from its porosity (higher VPD induces
reduction in pore size) under more aerosol scenario. India's mean temperature is constantly
rising [Krishnan et al., 2020]. At the same time, the global mean VPD is increasing with global
warming [Yuan et al., 2019] and heatwaves will be more frequent in future India [Mukherjee et
al., 2018]. Moreover, anthropogenic emissions over Indian Subcontinent will ensure high AOD
values in near future [Kumar et al 2018], thus manifesting a HAHT-like scenario at longer time
scales over India. Although, the response of plants and crops to enhancement in VPD in warmer
future is uncertain, but aerosol-induced weakening of VPD-EF associations can contribute
towards tendency of crops and vegetations becoming less drought/heat-resilient in future.

**6. Summary**

In summary, simultaneous observations from AERONET and an eddy covariance flux tower equipped with micrometeorological and soil physics sensors were employed to report possible influence of aerosol loading on VPD-Evaporative Fraction associations over a natural C4 grassland site under clear sky conditions in the central Gangetic Plains. The main findings from this study are:

1. Increase in aerosol loading reduces the incoming solar radiation at surface and reduces the gradient between surface temperature and near-surface air temperature. This is associated with the decrease in energy dissipation from surface via sensible heat. At the same time, increase in aerosol loading increases the evapotranspiration efficiency of ecosystem by increasing diffuse radiation. Thus, high aerosol loading favors dissipation of available surface energy via Latent heat flux and therefore increases Evaporative fraction.

2. Increase in surface temperature and VPD during heatwave conditions induce larger canopy resistance and stomata closure, thereby reducing the LH fluxes and EF. Native Plants tend to store more water by transpiring less in high temperature conditions; so GPP (and thus LH) reduces under high temperatures. At the same time, higher air temperature, also reduces the sensible heat partitioning via reduction in near surface temperature gradient. Thus, as the effect of VPD involves reducing both the surface fluxes, the net effect on EF is negligible.

3. The variability in aerosol loading tends to play a significant role in modulating the VPD-EF association under varying VPD/surface temperature. When the changes in VPD and scattering aerosols are in phase, like in case of stagnant heat wave conditions over North India, the VPD-induced reduction in evapotranspiration may be completely compensated. This physiological changes can be due to the aerosol fertilization effect or thick aerosol deposition/coating on leaves. Besides, as both increasing AOD and $T_{air}$ induces suppression in sensible heat partitioning, largely the changes in net EF remains in phase with changes in AOD and VPD.


Nonetheless, a few caveats of this study need to be kept in mind. Our analysis, although driven
by fundamental theory of land-atmosphere interactions, is statistical in nature with a relatively
small sample size. The cases we analyse here and carefully selected to represent the distinct
scenarios as far as realistically possible in this region. Thus, minor influences of meteorological
co-variability cannot be totally avoided. As such, the quantitative estimation of various
associations may have inherent uncertainties and care should be taken before generalizing.
Moreover, as literature on plant physiological responses specific to grass variants found in the
Indo-Gangetic Basin region are scare, this study warrants more species-level studies are
necessary to isolate the physiological and environmental responses on EF. Nevertheless, the
possible AOD-VPD-EF associations discussed here can have substantial implications on future
climate of this and similar subtropical regions. Thus, the observational associations provided in
this study not only encourages more measurements, detailed in situ experiments and mechanistic
modelling of aerosol-vegetation-atmosphere interactions, but also warrants proper
representations of aerosol processes and feedbacks in coupled models over India.

**Acknowledgement:**
SNT gratefully acknowledge the financial support given by the Earth System Science
Organization, Ministry of Earth Sciences, Government of India (grant MM/NERC-MoES-
03/2014/002) and Newton Fund to conduct this research under Monsoon Mission. CS
acknowledges support from MHRD, India under project number SB20210835CEMHRD00850.
LMM acknowledges the support of the Natural Environment Research Council (NERC) South
AMerican Biomass Burning Analysis (SAMBBA) project grant code NE/J010057/1. The authors
would like to thank Dr E. J. Welton, B.N. Holben and staff at NASA GSFC for establishing and
quality control of the AERONET and MPLNET site at IIT Kanpur, used in this study.

**Data statement:**

Surface data used here is available at: https://catalogue.ceh.ac.uk/documents/78c64025-1f8d-
431c-bdeb-e69a5877d2ed. Aerosol data used here is available from
https://www.iitk.ac.in/ce/aeronet.

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

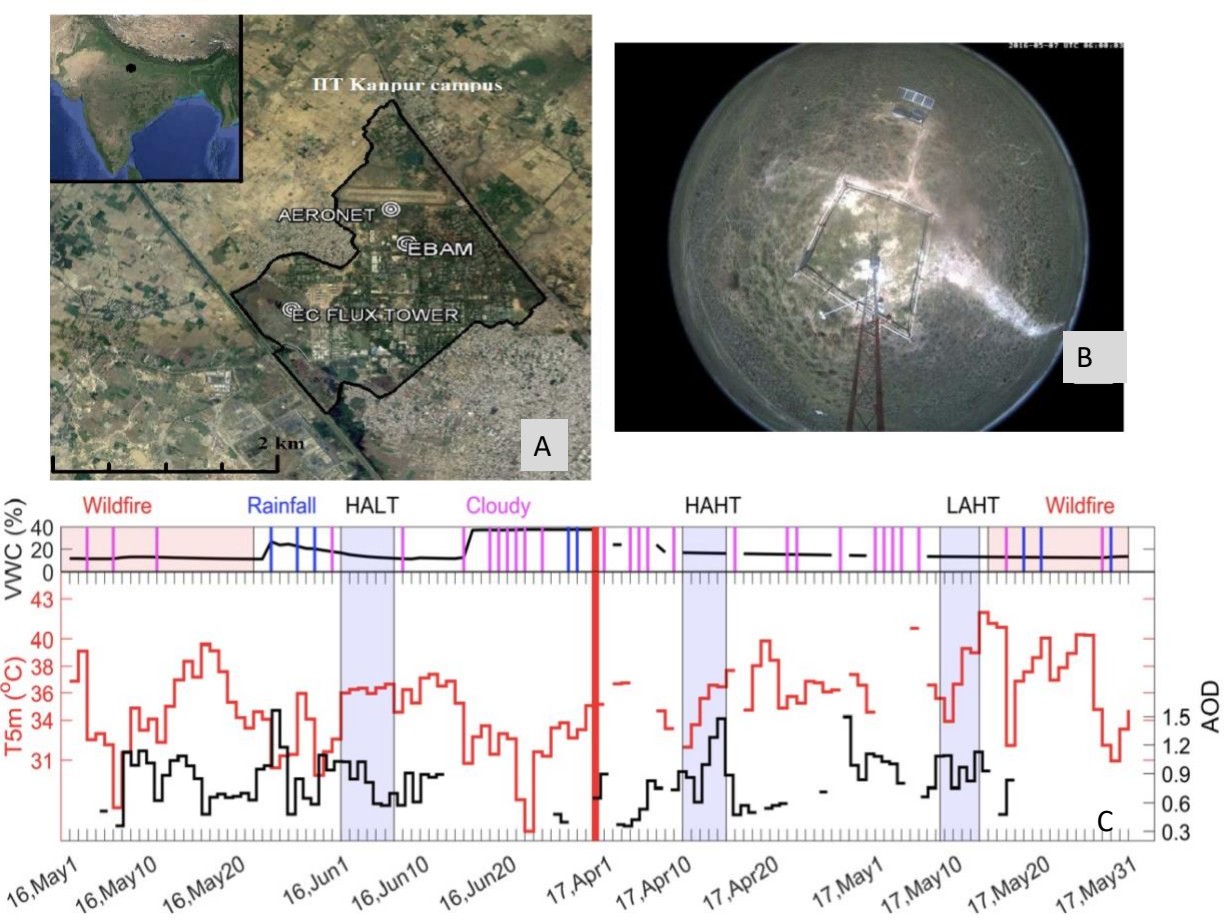


*Figure 1: A) Map showing the locations of AERONET and the EC flux tower site within the*
*campus of the Indian Institute of Technology Kanpur (IITK). Inset map shows the location of*
*IITK (black dot) in the central Gangetic Plains. The maps are created by © Google Maps 2017.*
*B) Camera image of land cover of the flux tower site during May 12th, 2017. C) Daily variation*
*in soil moisture (VWC, volumetric water content) during our study period is shown in black line*
*in upper box of the figure. The occurrences of cloudy days, rainy days and wildfire affected*
*period during April through June of 2016 and 2017 is shown by magenta, blue and pink colour*
*patches in the upper box. A cloudy day is inferred from MPLNET images and AERONET*
*observations (as defined in Section 2 of main text).  The days bounded by straight lines depict the*
*weekly episodes HALT, HAHT and LAHT, respectively. Daily variation in $T_{air}$ and daily*
*variation in AOD during our study period is shown as black and red lines in lower box of the*
*panel.*

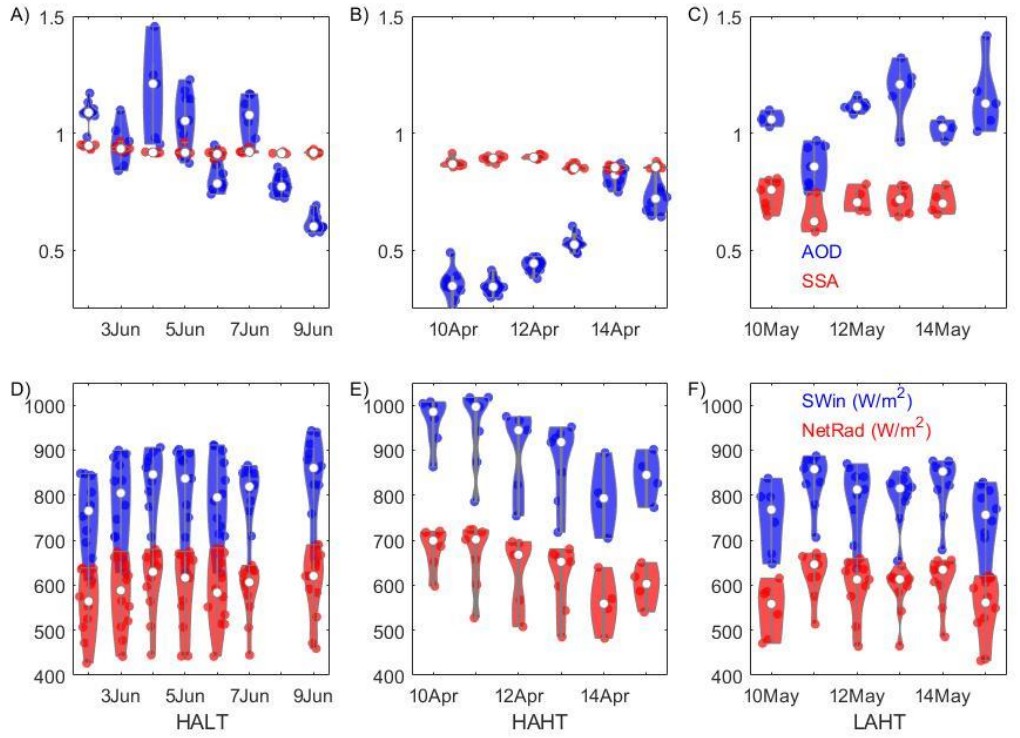

———————————————
*Figure 2: Distribution plots showing the variations in aerosol and radiation during the cases.*
*Row 1 illustrates Time series of midday (1100-1400 LT) variation in AOD and SSA values*
*during HALT, HAHT and LAHT, respectively.. The horizontal line within box represents median*
*of the distribution. The bottom and top edge of the boxes represent 25th and 75th percentile,*
*respectively, of the distribution. The short dash at top and bottom extent of the boxes represent*
*5th and 95th percentile, respectively. Row 2 is same as Row 1 but show measurements of*
*incoming short wave radiation and net radiation at surface. Note that June,16 means June of*
*2016 and so on.*

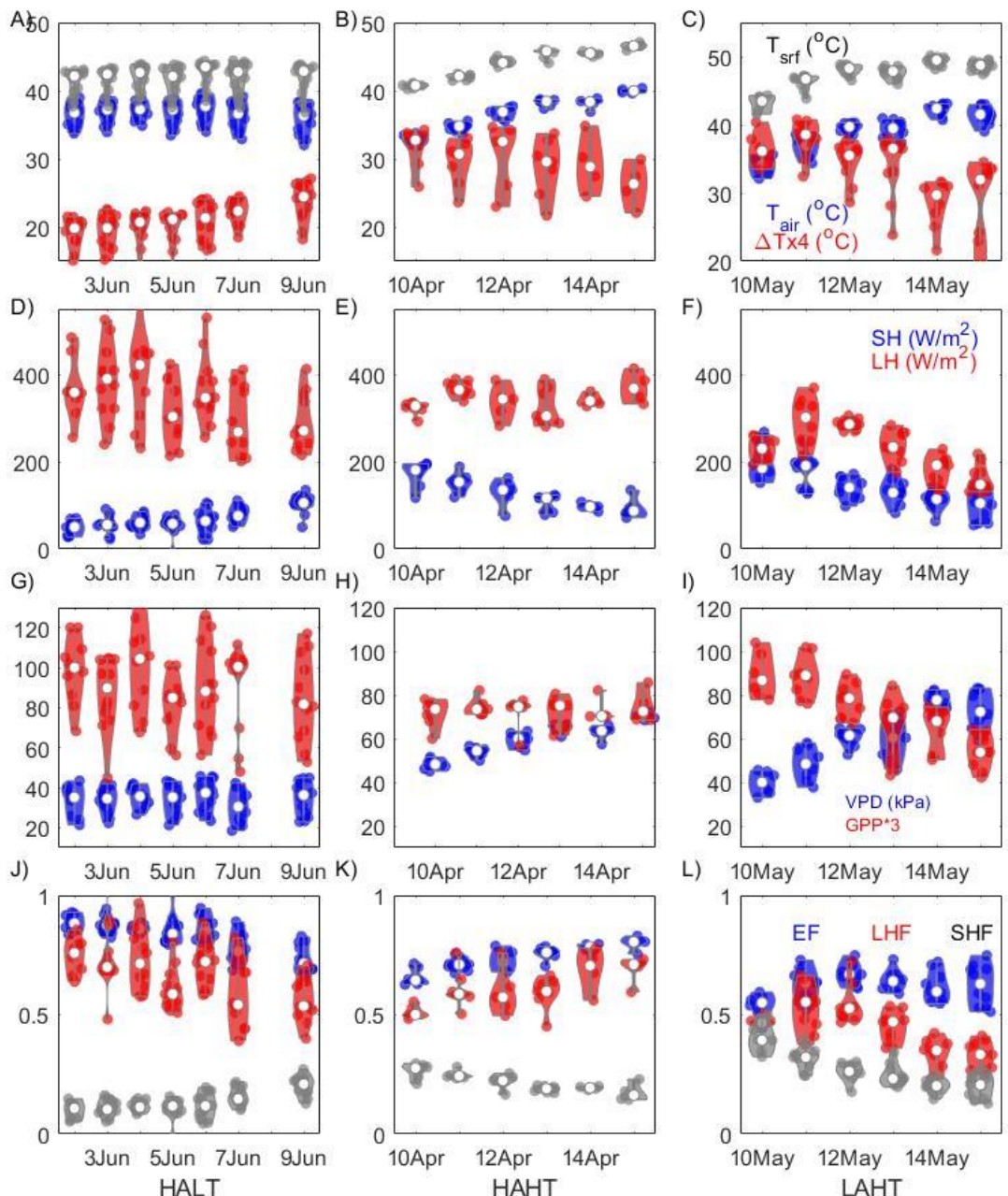


*Figure 3: Distribution plots showing the variations in near surface meteorology and surface*
*fluxes during the cases. Row 1 illustrates Time series of midday (1100-1400 LT) variation in $T_{srf}$,*
*$T_{air}$ and (-)$\Delta T$ values during HALT, HAHT and LAHT, respectively. Row 2 is same as Row 1 but*
*for SH and LH. Row 3 is same but for VPD and GPP ; Row 4 is same but for EF, LHF (red) and*
*SHF.*

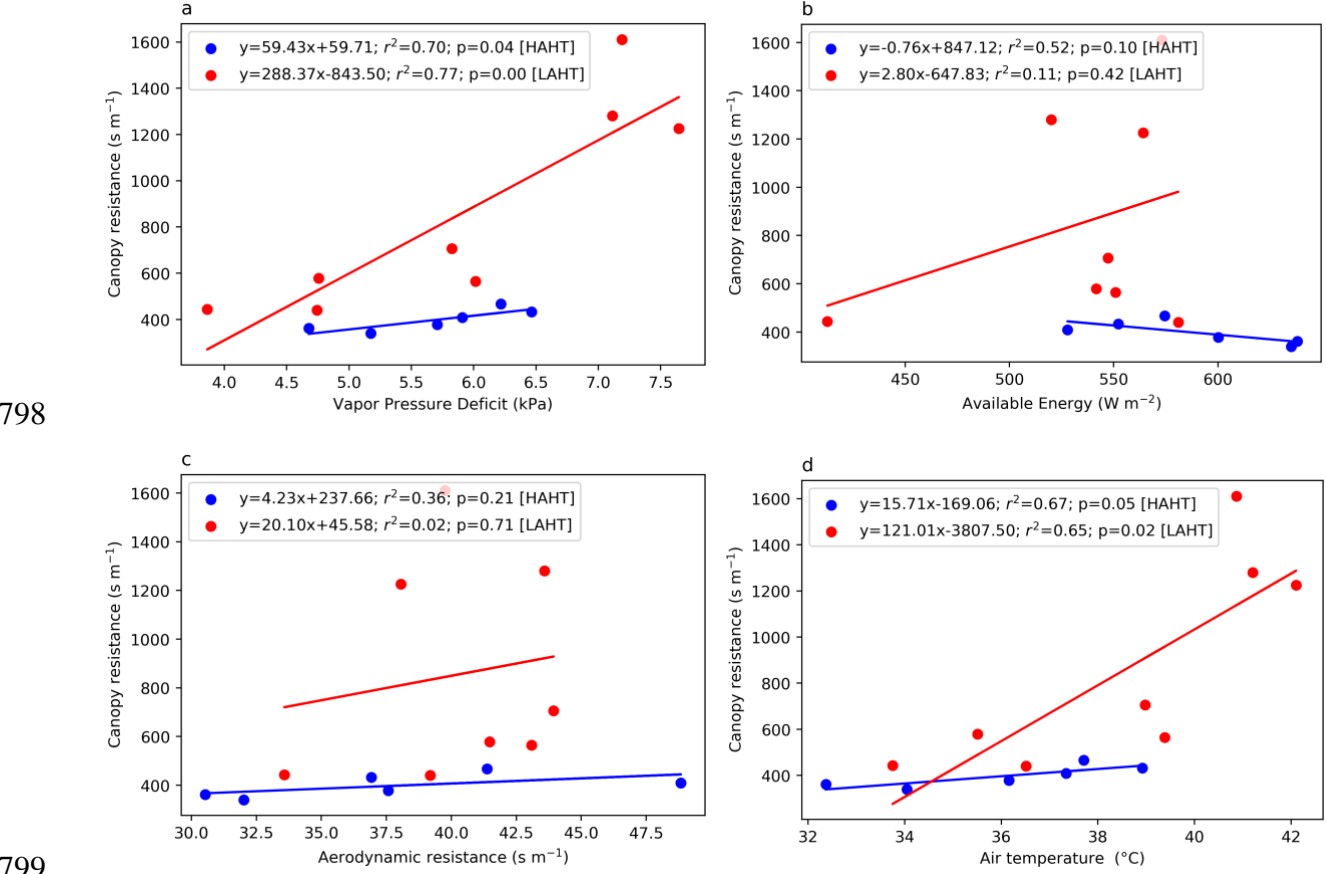



*Figure 4: Linear correlation between daily midday average Canopy resistance derived from*
*Penman-Monteith equation with a) observed Vapor Pressure Deficit (VPD); b) Available energy*
*at surface; c) Aerodynamic resistance and d) Air temperature for HAHT and LAHT cases.*













## Appendix A: Table of Abbreviations

| Name | Abrv. used |
|---|---|
| | |
| Latent heat flux | LH |
| Sensible heat flux | SH |
| Ground heat flux | GH |
| Evaporative Fraction | EF |
| 2 m air temperature | $T_{air}$ |
| vapor pressure deficit | VPD |
| gross primary production | GPP |
| net radiation | NR |
| aerosol direct radiative effect | ADRE |
| aerosol diffuse radiation fertilization effect | ADFE |
| diffuse radiation | $diffuse_{frac}$ |
| Santa Barbara discrete ordinates radiative transfer Atmospheric Radiative Transfer Model | SBDART |
| AErosol RObotic NETwork | AERONET |
| Volumetric soil water content | VWC |
| surface temperature | $T_{srf}$ |
| relative humidity | RH |
| Aerosol Optical Depth | AOD |
| Single Scattering Albedo | SSA |
| High AOD-Low $T_{air}$ | HALT |
| High AOD-High $T_{air}$ | HAHT |
| Low AOD- High $T_{air}$ | LAHT |
| Outgoing long wave radiation at surface | $LW_{out}$ |
| canopy resistance | $r_s$ |
| aerodynamic resistance to heat transfer | $r_a$ |
| Sensible heat fraction | SHF |
| Latent heat fraction | LHF |
