# Peer review of "Supporting Information for"

_Atmospheric Chemistry and Physics, 2021_

## Author Comment (AC1)

Reviewer 1

This manuscript addresses modeling aspects of plant atmosphere coupling and the role aerosols might play in it, with a focus on situations during the onset of the Indian summer monsoon. This is not a mechanistic study and thus has to rely on covariances. For doing this, it makes use of three typical situations that are classified by the variations of aerosol concentrations and temperatures. While I think this is generally a valid idea and could work, I also see a considerable amount of uncertainties and unclear definitions which reduce the validity.

We are thankful to the reviewer for appreciating the idea and approach of this study. Our point-by-point response for the comments are provided below in **blue** font. The new text/figure added to the revised manuscript is also provided in *italics* font.

We do acknowledge that due to the statistical nature of the analysis, it is difficult to draw quantitative and mechanistic conclusions. However, these observations provide new perspective and insights into atmosphere-biosphere interactions over this region that can inform the uncertainties in the current models, furthering fundamental research, and model development. In this context, we have added the following paragraph on the limitations of this study in summary section now:

*Added/Modified text in revised MS:*

*Lines 496-510*

*Nonetheless, a few caveats of this study need to be kept in mind. Our analysis, although driven by fundamental theory of land-atmosphere interactions, is statistical in nature with a relatively small sample size. The cases we analyse here and carefully selected to represent the distinct scenarios as far as realistically possible in this region. Thus, minor influences of meteorological co-variability cannot be totally avoided. As such, the quantitative estimation of various associations may have inherent uncertainties and care should be taken before generalizing. Moreover, as literature on plant physiological responses specific to grass variants found in the Indo-Gangetic Basin region are scare, this study warrants more species-level studies are necessary to isolate the physiological and environmental responses on EF. Nevertheless, the possible AOD-VPD-EF associations discussed here can have substantial implications on future climate of this and similar subtropical regions.*

Q. First, the examples are not so well chosen. In Fig. 1, the difference in AOD variation during the HALT period and the LAHT period is not clear. The distance between AOD minimum and AOD maximum during both periods is almost the same, just the mean trend is different (decreasing during HALT; increasing during LAHT). Why is the fourth possible scenario (LALT), not mentioned, would it support the conclusions?

Figure 1 shows the variability in temperature or VPD and AOD during the campaign. The three one-week cases are specially selected such that the data analysis can help to understand the AOD-VPD-EF association under ambient conditions observed at our site. Moreover, we also calculated the aerosol direct effect and aerosol-induced changes in diffused radiation (aerosol fertilization effect) for the three weeks (shown in Figure S1). This informs on the variation in aerosol effect on surface fluxes. We have selected the cases to have different combinations of weekly gradients (in AOD and T). This is because eventually we are

analyzing the weekly gradients in surface fluxes and EF and trying to understand the relative dominance on both the factors on EF. Note that during the three study weeks all other potential factors affecting EF have negligible variability.

Case 1)The HAHT week had high gradient in AOD and T across one week, both in phase, hence the weekly gradient in EF is assumed to be net effect of aerosol and VPD gradient; Case 2) The HALT week had high gradient in AOD but corresponding gradient in T was negligible, hence the gradient seen in EF during that week can be due to variability in aerosol-EF interactions. Both the above weeks also witnessed strong gradients in aerosol direct effect at surface and diffused radiation. Case 3) The LAHT case (10$^{th}$ to 15$^{th}$ May 2017) witnessed high gradient in T across the week, but no clear gradient in AOD i.e largely, the AOD values were close to 1. In the revised MS, we have included high resolution diurnal plots (shown below) which better clarifies these details. In agreement, negligible variations were also seen in aerosol direct effect and diffused radiation across the week. We have modified the text in many locations for better clarity in this context in revised MS (shown below). The major changes are included below.

During our study period, there was not a period when weekly gradients in both AOD and temperature are low, hence we have not mentioned/analysed the LALT scenario.

*Added/Modified text in revised MS:*

*Lines 202-223*

*Eventually, three one-week periods are carefully selected with different combinations of dominant weekly gradients in T$_{air}$ /VPD and AOD and analysed to gain insights into ambient AOD-VPD-EF association. The first week selected for analysis is between 2$^{nd}$-9$^{th}$ June, 2016, which had high weekly gradient in AOD but was accompanied by low variation in T$_{air}$/VPD (hereafter referred as High AOD-Low T$_{air}$ (HALT) case). The second week is during 10$^{th}$-15$^{th}$ April, 2017, which witnessed large daily increase in aerosol loading as well as T$_{air}$ in phase throughout the week (hereafter referred to as the High AOD-High T$_{air}$ (HAHT) case). We also selected a third week during 10$^{th}$-15$^{th}$ May, 2017, when high gradient in T$_{air}$ was observed across the week, but negligible weekly gradient in AOD was present i.e the AOD values had large day to day variability through the week (hereafter referred to as the Low AOD- High T$_{air}$ (LAHT) case). Interestingly, heatwave conditions were prevalent over North India during the HAHT and LAHT weeks, therefore, a wide range of VPD-AOD-EF variation can be sampled. Moreover, since there were no rainfall events during these three weeks, the variation in VWC was minor compared to large daily variations in T$_{air}$ and AOD during our study periods. Further, the variations in the vegetation phenology, wind and boundary layer height are found to be* negligible *within each of these three weeks. Note that no week with low AOD and low VPD variations was observed during our study period.*

*The simultaneous midday (1000-1500 LT) variability in AOD, VPD, EF and the other components of the surface radiative balance is analyzed across the HALT and LAHT weeks to understand the impact of strong weekly gradients of AOD and VPD, respectively. Further, we analyse the weekly gradients in the observations during HAHT, and compare and contrast the same with the HALT and LAHT cases to understand the combined effects of AOD and VPD.*

*Line 273-280*

*In contrast, during LAHT week, the gradient of AOD values from 10$^{th}$ and 15$^{th}$ May,17 was relatively minor (Figure 2C). As the increase in AOD through the week was smaller compared to other two cases, corresponding decrease of NR and ISWR values at surface was also smaller in magnitude (~30 W/m$^2$) during this period (Figure 2F). Correspondingly, negligible trend in ADRF (Figures S1C) at the surface is observed indicating low variation in aerosol radiative effect change during the LAHT week.*

[Figure]

Figure 1c: *Daily variation in soil moisture (VWC, volumetric water content) during our study period is shown in black line in upper box of the figure. The occurrences of cloudy days, rainy days and wildfire affected period during April through June of 2016 and 2017 is shown by magenta, blue and pink colour patches in the upper box. A cloudy day is inferred from MPLNET images and AERONET observations (as defined in Section 2 of main text). The days bounded by straight lines depict the weekly episodes HALT, HAHT and LAHT, respectively. Daily variation in T$_{air}$ and daily variation in AOD during our study period is shown as black and red lines in lower box of the panel.*

[Figure]

Figure R1: Diurnal evolution of AOD for the three case studies; HALT (top), HAHT (middle) and LAHT (bottom).

Q. Also, plants have different water strategies which particularly determines their response to vpd. Isohydric plants readily reduce stomatal aperture with increasing vpd, as it is assumed here. However, anisohydric plants tend to keep stomata open, some of them to the extent that they (nearly) become wilty, for the benefit of keeping up CO2 uptake and photosynthesis. What kind of strategy did the plants on the respective grassland use? Details about species are not given, apart from a semi-natural grassland with different C4 grasses representative for grasslands of the region. C4 grasses may be isohydric or anisohydric (e.g., Jardine, Thomas & Osborne, Ecology and Evolution, 2021), bringing the transpiration /EF reaction to vpd and the conclusions drawn in the manuscript into question. This point is my major criticism, as it can question the whole approach, si it must be considered.

This is a good point about the plant physiology specific to the Indo-Gangetic Basin region, however related studies are rare. Since the eddy covariance measurements are for a footprint around the flux tower, what we see here is the bulk behavior of C4 grasses in our site. Primarily, the C4 type grasses dominating our site are variants of Napier grass and some common reed (scientific family: *Pennisetum purpureum* and *Phragmites-Saccharum-Imperata*). Of these, Napier grasses dominate and is an invasive and perennial species (Holm et al., 1979). Napier grasses can be anisohydric, i.e. water spending under ample water availability (Cardoso et al., 2015). But their behavior becomes isohydric under high water stress and high temperature (Liang et al., 2017; Mwendia et al. 2014; Purbajanti et al., 2012). Reductions in Napier grass canopy conductance (or, in other words, increase in canopy resistance), with increasing VPD has also been observed in other studies (Heerwaarden and Teuling, 2016; Mwendia et al. 2016). In both HAHT and LAHT cases, the temperature enhancement is high and the soil moisture is very low. Hence, the water-stressed grasses

close their stomata with high VPD as supported by the regression slope between canopy resistance and VPD (Figure 4). However, the strong gradient of increase in canopy resistance with VPD/ air temperature gets moderated under high aerosol scenario indicating decoupling of canopy resistance from VPD changes. One possible mechanism for this is the diffuse radiation fertilization effect, which allows normally shaded leaves to increase their photosynthesis (and thus increase transpiration), effectively reducing the canopy resistance. Also, deposition of aerosols on leaves as wax can also impact the stomata behaviour. Recently, Grantz *et al*. 2018 used direct observations in glasshouses to illustrate decoupling of stomata conductance (flux-based) from its porosity (higher VPD induces reduction in pore size) under a high aerosol scenario.

However, VPD is not the only controlling factor of variations in canopy resistance. The interactions between multiple factors, including available energy, temperature, and VPD, control the canopy resistance. Hence, we have investigated the relationships between other factors that control evapotranspiration, namely available energy and moisture demand (the physical factors) and the aerodynamic resistance (a physiological factor) and canopy resistance during the LAHT and HAHT cases (Fig 4cb-d). We have also checked for statistical significant of these relationships. We find that the canopy resistance is only significantly (p<0.05) correlated with VPD, but not with available energy and the aerodynamic resistance. Additionally, the sensitivity of canopy resistance to changes in VPD is much higher than that for the available energy and the aerodynamic resistance. Similarly, increase in air temperature during these periods also shows statistically significant positive relationships with canopy resistance, which is consistent with our understanding that Napier grasses under high water stress close their stomata at high ambient temperature (Heerwaarden and Teuling, 2016). These additional panels are now added to Figure 4 (shown below).

*Modified text:*

*Line 145-148*

*The fetch around the tower is a mixture of different C4 grasses, i.e. variants of Napier grass and some common reed (Scientific family: Pennisetum purpureum and Phragmites-Saccharum-Imperata). Napier grasses are invasive and a perennial species and representative of grasslands in the region (Chakraborty et al., 2019; Holm et al., 1979).*

*Line 355-383:*

> *Figure 4 illustrates the variation in midday mean canopy resistance during the LAHT and HAHT weeks to various physical and physiological factors that control evapotranspiration, namely moisture demand, available energy, air temperature and the aerodynamic resistance. As expected, the canopy resistance is significantly (p<0.05) correlated with VPD although clear differences in the slope is present for the two cases. Specifically, the canopy resistance increases steeply from 400 to 1400 s m$^{-1}$ with increase in VPD from 40 to 70 hPa during LAHT case (Figure 4a). However, the canopy resistance only increases from 400 to 500 with an increase in VPD from 45 to 65 hPa during HAHT case (Figure 4a). Similarly, air temperature during these periods also shows a statistically significant positive relationship with canopy resistance (Figure 4d). However, during both periods, canopy resistance was found to be independent of available energy (Figure 4c) and the aerodynamic resistance (Figure 4d), indicating that the sensitivity of canopy resistance to changes in VPD (or $T_{air}$ ) is significantly greater than that for the other variables.*

The LAHT case illustrates the frequently reported behaviour of reduction of canopy conductance under increasing VPD due to partial stomata closure as a physiological stress response (Grossiord et al., 2020). Similar responses are also reported in Napier grasses, the native vegetation over our site (Mwendia et al. 2016). Napier grasses can be anisohydric, i.e. water spending under ample water availability (Cardoso et al., 2015). But their behaviour becomes isohydric under high temperature and high water stress (Liang et al., 2017; Mwendia et al. 2014; Purbajanti et al., 2012). During both HAHT and LAHT weeks, soil moisture is very low, hence, the Napier grasses behaves isohydrically under high VPD. The comparison of LAHT and HAHT scatter illustrates that canopy conductance is not strongly affected even under severe VPD rise when aerosol loading also increases in phase. Specifically, the strong gradient of increase in canopy resistance with VPD/ air temperature gets moderated under the high aerosol scenario. Thus, under the presence of high aerosol loading, the isohydric response of Napier grass to temperature rise or the physiological stress under high VPD is decoupled. This can partially explain the aerosol-induced increase in EF (as well as LH and GPP) even under high VPD rise during HAHT.

Line 505-513:

Our analysis, although driven by fundamental theory of land-atmosphere interactions, is statistical in nature with a relatively small sample size. The cases we analyse here were carefully selected to represent the distinct conditions in this region. Thus, minor influences of meteorological co-variability cannot be totally avoided. As such, the quantitative estimation of various associations may have inherent uncertainties and care should be taken before generalizing. Moreover, as literature on plant physiological responses specific to grass variants found in the Indo-Gangetic Basin region are scarce, this study warrants more species-level studies are necessary to isolate the physiology and environmental responses on EF. Nevertheless, the possible AOD-VPD-EF associations discussed here can have substantial implications on future climate of this and similar subtropical regions.

[Figure]

*Figure 4: Linear correlations between daily midday average Canopy resistance derived from Penman-Monteith equation with a) observed Vapor Pressure Deficit (VPD); b) Available energy at surface; c) Aerodynamic resistance and d) Air temperature for HAHT and LAHT cases.*

Q. The manuscript is very difficult to read. It should include a table with explanations for the more than 30 abbreviations used. These are too many for keeping all in mind and going back to the first mention is impractical.

Thank you for this suggestion, we have now included a table of abbreviations used as Appendix A in the revised manuscript for the ease of readers.

**Appendix A: Table of Abbreviations**

| Name | Abrv. used |
|---|---|
| | |
| Latent heat flux | LH |
| Sensible heat flux | SH |
| Ground heat flux | GH |
| Evaporative Fraction | EF |
| 2 m air temperature | $T_{air}$ |
| vapor pressure deficit | VPD |
| gross primary production | GPP |
| net radiation | NR |
| aerosol direct radiative effect | ADRE |
| aerosol diffuse radiation fertilization effect | ADFE |
| diffuse radiation | $diffuse_{frac}$ |
| Santa Barbara discrete ordinates radiative transfer Atmospheric Radiative Transfer Model | SBDART |
| AErosol RObotic NETwork | AERONET |
| Volumetric soil water content | VWC |
| surface temperature | $T_{srf}$ |
| relative humidity | RH |
| Aerosol Optical Depth | AOD |
| Single Scattering Albedo | SSA |
| High AOD-Low $T_{air}$ | HALT |
| High AOD-High $T_{air}$ | HAHT |
| Low AOD- High $T_{air}$ | LAHT |
| Outgoing long wave radiation at surface | $LW_{out}$ |
| canopy resistance | $r_s$ |
| aerodynamic resistance to heat transfer | $r_a$ |
| Sensible heat fraction | SHF |
| Latent heat fraction | LHF |

Q. What is more, the manuscript lacks thorough definitions. The word 'continuum' is used as ,Aerosol-plant-temperature-EF continuum' (l. 36), as ,'land-atmosphere-energy balance continuum' (194), and as 'aerosol-Tair-VPD-EF continuum' (l. 426). A thorough definition of a continuum would be something as the soil-plant-atmosphere continuum (SPAC), an established term in plant physiology, based on the water potential as a driving, unifying factor that determines the flow of water and water vapor, independent of the physical water status (Liquid water or water vapor). Maybe something like connection is meant here, but it is really difficult to guess.

We are thankful to the reviewer for pointing this out.
We have revised the manuscript thoroughly to only use clear definitions.
Specifically,
Aerosol-plant-temperature-EF continuum is replaced by 'AOD-VPD-EF coupling'. Land-atmosphere-energy balance continuum in line 194 is removed from revised MS.
Aerosol-Tair-VPD-EF continuum is replaced by 'AOD-VPD-EF coupling'.

**References:**
1. Mwendia, S. W., Yunusa, I. A., Sindel, B. M., Whalley, R. D., & Kariuki, I. W. (2017). Assessment of Napier grass accessions in lowland and highland tropical environments of East Africa: water stress indices, water use and water use efficiency. *Journal of the Science of Food and Agriculture*, *97*(6), 1953-1961.
2. van Heerwaarden, C. C., & Teuling, A. J. (2014). Disentangling the response of forest and grassland energy exchange to heatwaves under idealized land–atmosphere coupling. *Biogeosciences*, *11*(21), 6159-6171.
3. Chakraborty, T., Sarangi, C., Krishnan, M., Tripathi, S. N., Morrison, R., & Evans, J. (2019). Biases in model-simulated surface energy fluxes during the Indian monsoon onset period. *Boundary-Layer Meteorology*, *170*(2), 323-348.
4. Juan Andrés Cardoso, Marcela Pineda, Juan de la Cruz Jiménez, Manuel Fernando Vergara, Idupulapati M. Rao, Contrasting strategies to cope with drought conditions by two tropical forage C$_4$grasses, *AoB PLANTS*, Volume 7, 2015, plv107, https://doi.org/10.1093/aobpla/plv107
5. Liang, X., Erickson, J.E., Sollenberger, L.E., Rowland, D.L., Silveira, M.L. and Vermerris, W. (2018), Growth and Transpiration Responses of Elephantgrass and Energycane to Soil Drying. Crop Science, 58: 354-363. https://doi.org/10.2135/cropsci2017.01.0019
6. Purbajanti, E.; Anwar, S.; Wydiati, F.K. Drought stress effect on morphology characters, water use efficiency, growth and yield of guinea and napier grasses. *Int. Res. J. Plant Sci.* 2012, *3*, 47. [Google Scholar]
7. Mwendia, S.; Yunusa, I.; Whalley, R.; Sindel, B.; Kenney, D.; Kariuki, I. Use of plant water relations to assess forage quality and growth for two cultivars of Napier grass (*Pennisetum purpureum*) subjected to different levels of soil water supply and temperature regimes. *Crop Pasture Sci.* 2014, *64*, 1008–1019. [Google Scholar]
8. Holm L, Pancho JV, Herberger JP, Plucknett DL, 1979. A Geographical Atlas of World Weeds. Toronto, Canada: John Wiley and Sons Inc
9. Charlotte Grossiord, Thomas N. Buckley, Lucas A. Cernusak, Kimberly A. Novick, Benjamin Poulter, Rolf T. W. Siegwolf, John S. Sperry, Nate G. McDowell
10. Irmak, S., and Mutiibwa, D. (2010), On the dynamics of canopy resistance: Generalized linear estimation and relationships with primary micrometeorological variables, *Water Resour. Res.*, 46, W08526, doi:10.1029/2009WR008484.

---

## Author Comment (AC2)

Reviewer 2:

This paper is an interesting discussion of the link between aerosols, vapor pressure deficit and evapotranspiration over India. The paper presents some interesting findings: 1) sensible heat is lower under heat wave conditions, 2) latent heat is enhanced under aerosol loading due to diffuse fertilization, and 3) decoupling of the vapor pressure deficit response under high aerosol load. These are very interesting findings, as they turn out to be different than what is common knowledge for regions that do not have the aerosol load of India and provides insights into the coupled behavior of air pollution, vegetation, and weather.

We thank the reviewer for his appreciation of the interesting finding of our study. Our point-to-point response to all the comments are provided below in blue font and the corresponding modification in the revised MS is shown in *italics*.

Major comments:

Q. The finding that the evaporation response to vapor pressure deficit becomes really weak under a high aerosol optical depth is very interesting, but also controversial. The authors demonstrate the opposite findings in a modelling study, which shows that their results might be very important. At the same time: one figure (Fig. 4) does not really convince me. The explanation of it remains rather limited and I think that this finding deserves a far more thorough analysis before this paper can be accepted. Vapor pressure deficit is not the only driver of stomatal resistance, and it would be good to carefully look into each of them. It would be nice to analyze here a few diurnal cycles into detail. I would like to see the evolution of the evapotranspiration and specific humidity, next to the radiation and the surface temperature.

We do acknowledge that due to the statistical nature of the analysis, it is difficult to draw clear mechanistic conclusions. However, these unprecedented set of observations provide a good platform to analyze and gain insights into role of aerosols on VPD-EF associations that can inform further research and model development.

VPD is not only controlling factor for canopy resistance. The interactions between multiple factors, including available energy, temperature, and VPD, control the canopy resistance. Hence, we have investigated the relationships between other factors that control evapotranspiration, namely available energy and moisture demand (the physical factors) and the aerodynamic resistance (a physiological factor) and canopy resistance during the LAHT and HAHT cases (Fig 4cb-d). We have also checked for statistical significant of these relationships. We find that the canopy resistance is only significantly ($p<0.05$) correlated with VPD, not the other two variables. Additionally, the sensitivity of canopy resistance to changes in VPD is much higher than that for the other two variables. Similarly, increase in air temperature during these periods also show statistically significant positive relationships with canopy resistance, which is consistent with our understanding that plants close their stoma at high ambient temperature (Heerwaarden and Teuling, 2016). These additional panels are now added to Figure 4. Also see them below.

We also analysed the diurnal evolution of micro-meteorological variables such as soil temperature and moisture, specific humidity, incoming solar radiation along with latent heat, GPP and $CO_2$ fluxex. As heatwave was prevalent during HAHT and LAHT weeks with substantial increases in soil temperature, which resulted in minor decrease in soil moisture across both weeks. Moreover, some variations are seen in the evolution of wind speed during HAHT as it decreased by ~3-4 m/s from 10[th] April to 15[th] April, 2017 during HAHT. All other meteorological variables showed negligible weekly trends during HAHT.

Largely, evapotranspiration is expected to vary proportionally with wind speed, if all other factors remain same, however we find that both GPP and latent heat, increase gradually during HAHT, indicating secondary/tertiary impact of wind speed variation on evapotranspiration during this week.

During LAHT, all the meteorological variables also showed negligible temporal trends except specific humidity. The specific humidity decreased from 10[th]May to 15[th] May, 2017, which is similar to the decreasing trend in evapotranspiration. The consistency could be probably because evapotranspiration is a main source of near surface moisture over our site during stagnant heat wave conditions in dry season. Thus, a closer look illustrates that although minor gradients are present in the meteorological variables, they are not dominant factors influencing evapotranspiration variation during the two case studies. Nonetheless, the individual or relative contribution of these meteorological variability and aerosols on the observed coupling deserves further attention in future studies with in depth mechanistic modelling.

*Modified text:*

*Line nos: 355-395*

*Figure 4 illustrates the variation in midday mean canopy resistance during the LAHT and HAHT week to various physical and physiological factors that control evapotranspiration, namely moisture demand, available energy, air temperature and the aerodynamic resistance. As expected, the canopy resistance is significantly (p<0.05) correlated with VPD although clear differences in the slope is present for the two cases. Specifically, the canopy resistance increases steeply from 400 to 1400 s m$^{-1}$ with increase in VPD from 40 to 70 hPa during LAHT case (Figure 4a). However, the canopy resistance increases from 400 to 500 with increase in VPD from 45 to 65 hPa during HAHT case (Figure 4a). Similarly, increase in air temperature during these periods also show statistically significant positive relationships with canopy resistance (Figure 4d). However, during both the weeks, canopy resistance was found to be independent of available energy (Figure 4c) and the aerodynamic resistance (Figure 4d), indicating that the sensitivity of canopy resistance to changes in VPD or $T_{air}$ is significantly greater than that for the other variables.*

*The LAHT case illustrates the frequently reported behaviour of reduction of canopy conductance under increase in VPD due to partial stomata closure as a physiological stress response (Grossiord et al., 2020). Similar responses are also reported in Napier grasses, the native vegetation over our site (Mwendia et al. 2016). Napier grasses can be anisohydric, i.e. water spending under ample water availability (Cardoso et al., 2015). But their behavior becomes isohydric under high temperature and high water stress (Liang et al., 2017; Mwendia et al. 2014; Purbajanti et al., 2012). During both HAHT and LAHT weeks, the soil moisture is very*

*low. Hence, the Napier grasses behaves isohydric-ally under high VPD. Interestingly, the comparison of LAHT and HAHT scatter illustrates that canopy conductance is not much affected even under severe VPD rise when aerosol loading also increases in phase. Specifically, the strong gradient of increase in canopy resistance with VPD/ air temperature gets moderated under high aerosol scenario. Thus, under the presence of high aerosols loading, this isohydric response of Napier grass to temperature rise or the physiological stress under VPD increase is decoupled. This can partially explain the aerosol-induced increase in EF (as well as LH and GPP) even under high VPD rise during HAHT.*

*Further, meteorological co-variability or any significant differences in weekly pattern of other micro-meteorological variables between HAHT and LAHT cases can also contribute to the corresponding differences in AOD-VPD-EF association. A closer look illustrates that minor gradients are present in the meteorological variables (Figure S2), which can have secondary effects on the VPD-EF associations. Nonetheless, the individual or relative contribution of these meteorological variability and aerosols on the observed coupling remains unknown and deserves further attention in future studies with in depth mechanistic modelling.*

[Figure]

Figure 4: Linear correlation between daily midday average Canopy resistance derived from Penman-Monteith equation with a) observed Vapor Pressure Deficit (VPD); b) Available energy at surface; c) Aerodynamic resistance and d) Air temperature for HAHT and LAHT cases. Modified text

[Figure]

*Figure S2: The daily evolution of meteorological variables during LAHT and HAHT weeks.*

Q. The inversion of Penman-Monteith that leads to figure 4 is not reproducible. I would like to see this method thorougly described in the paper. Furthermore, I am a little skeptical of using

surface temperature here. Please also compute the stomatal resistance using the air temperature as Penman-Monteith does as well.

We have now included the complete methodology in the revised manuscript with the relevant equations. Note that surface temperature is only used to derive aerodynamic resistance using the observed sensible heat flux and near-surface temperature gradient.

*Modified text*

*Line 229*

*We also calculated the daily midday bulk canopy resistances for both HAHT and LAHT cases by inverting the Penmann-Monteith equation as described below. We used observed values of available energy, VPD, $T_{srf}$ derived from observed $LW_{out}$, psychrometric constant and slope of vapor pressure curve derived from observed surface pressure and $T_{air}$ respectively, and aerodynamic resistance derived from the observed SH and near-surface temperature gradient.*

*The aerodynamic resistance to heat transfer ($r_a$) is calculated from the near-surface temperature gradient and the measured H, given by:*

$$r_a = \frac{-\rho C p \,(Tsrf - Tair)}{H}$$

*where $T_s$ is the surface temperature, calculated by inverting the Stefan-Boltzmann law assuming a unit surface emissivity (reasonable for vegetated surfaces), $\rho$ is the air density, and $C_p$ is the specific heat at constant pressure (1.005 x 10$^{-3}$ MJ kg$^{-1}$ °C$^{-1}$).*

*Then, the canopy resistance ($r_s$) is calculated by inverting the Penman-Monteith approximation. Thus:*

$$r_s = \frac{\left(\dfrac{\Delta(Rn - G) + \dfrac{\rho C p VPD}{r_a}}{LE}\right) - \Delta}{\gamma - 1} r_a$$

*where $\Delta$ is the slope of the water vapor saturation curve given by:*

$$\Delta = \frac{4098\left[0.6108 exp\left(\frac{17.27 T_a}{T_a + 237.3}\right)\right]}{(T_a + 237.3)^2}$$

*and $\gamma$ is the psychrometric constant, calculated as:*

$$\gamma = \frac{C_p P}{\varepsilon \lambda}$$

*where P is atmospheric pressure in kPa, $\lambda$ is the latent heat of vaporization (2.45 MJ kg$^{-1}$), and $\varepsilon$ is the ratio of the molecular weight of water vapour to dry air (0.622).*

Minor comments:

* In my view, all acronyms could be replaced by written words in order to make the paper more readable. It does do no harm if the paper is 20 lines longer for that reason.

We have now expanded the abbreviations in most of the new paragraphs in the revised text for ease of readers. Moreover, we have added a table of abbreviations used for ease of the readers.

**Appendix A: Table of Abbreviations**

| Name | Abrv. used |
|---|---|
|  |  |
| Latent heat flux | LH |
| Sensible heat flux | SH |
| Ground heat flux | GH |
| Evaporative Fraction | EF |
| 2 m air temperature | $T_{air}$ |
| vapor pressure deficit | VPD |
| gross primary production | GPP |
| net radiation | NR |
| aerosol direct radiative effect | ADRE |
| aerosol diffuse radiation fertilization effect | ADFE |
| diffuse radiation | $diffuse_{frac}$ |
| Santa Barbara discrete ordinates radiative transfer Atmospheric Radiative Transfer Model | SBDART |
| AErosol RObotic NETwork | AERONET |
| Volumetric soil water content | VWC |
| surface temperature | $T_{srf}$ |
| relative humidity | RH |
| Aerosol Optical Depth | AOD |
| Single Scattering Albedo | SSA |
| High AOD-Low $T_{air}$ | HALT |
| High AOD-High $T_{air}$ | HAHT |
| Low AOD- High $T_{air}$ | LAHT |
| Outgoing long wave radiation at surface | $LW_{out}$ |
| canopy resistance | $r_s$ |
| aerodynamic resistance to heat transfer | $r_a$ |
| Sensible heat fraction | SHF |
| Latent heat fraction | LHF |

* The overall quality of the figures is too poor for publication. Please make sure all figures have a consistent font size, are not stretched and have either a vector format, or a high enough resolution.

All Figures are replotted and extracted at finer resolutions for improvement in clarity.

[Figure]

*Figure 2: Distribution plots showing the variations in aerosol and radiation during the cases. Row 1 illustrates Time series of midday (1100-1400 LT) variation in AOD and SSA values during HALT, HAHT and LAHT, respectively.. The horizontal line within box represents median of the distribution. The bottom and top edge of the boxes represent 25th and 75th percentile, respectively, of the distribution. The short dash at top and bottom extent of the boxes represent 5th and 95th percentile, respectively. Row 2 is same as Row 1 but show measurements of incoming short wave radiation and net radiation at surface. Note that June,16 means June of 2016 and so on.*

[Figure]

*Figure 3: Distribution plots showing the variations in near surface meteorology and surface fluxes during the cases. Row 1 illustrates Time series of midday (1100-1400 LT) variation in $T_{srf}$, $T_{air}$ and (-)$\Delta T$ values during HALT, HAHT and LAHT, respectively. Row 2 is same as Row 1 but for SH and LH. Row 3 is same but for VPD and GPP ; Row 4 is same but for EF, LHF (red) and SHF.*

\* Please use units consistently, I see W/m2 as well as W m^{-2}. Please add a space between different units.

Corrected.

\* Line 71-73: the paper of Van Heerwaarden & Teuling (2014, Biogeosciences) shows exactly the threshold where VPD increase leads to a shutdown, rather than increase in ET.

We thank the reviewer for the reference. Please see our response to your main comment#1 above, where we have included relevant discussion on this point.

\* Figure 4: Please check the units of VPD, these must be Pa for these values.

We have corrected this plot as below.

[Figure]

References:

1. Mwendia, S. W., Yunusa, I. A., Sindel, B. M., Whalley, R. D., & Kariuki, I. W. (2017). Assessment of Napier grass accessions in lowland and highland tropical environments of East Africa: water stress indices, water use and water use efficiency. *Journal of the Science of Food and Agriculture*, *97*(6), 1953-1961.
2. van Heerwaarden, C. C., & Teuling, A. J. (2014). Disentangling the response of forest and grassland energy exchange to heatwaves under idealized land–atmosphere coupling. *Biogeosciences*, *11*(21), 6159-6171.
3. Chakraborty, T., Sarangi, C., Krishnan, M., Tripathi, S. N., Morrison, R., & Evans, J. (2019). Biases in model-simulated surface energy fluxes during the Indian monsoon onset period. *Boundary-Layer Meteorology*, *170*(2), 323-348.
4. Juan Andrés Cardoso, Marcela Pineda, Juan de la Cruz Jiménez, Manuel Fernando Vergara, Idupulapati M. Rao, Contrasting strategies to cope with drought conditions by two tropical forage C4grasses, *AoB PLANTS*, Volume 7, 2015, plv107, https://doi.org/10.1093/aobpla/plv107

5. Liang, X., Erickson, J.E., Sollenberger, L.E., Rowland, D.L., Silveira, M.L. and Vermerris, W. (2018), Growth and Transpiration Responses of Elephantgrass and Energycane to Soil Drying. Crop Science, 58: 354-363. https://doi.org/10.2135/cropsci2017.01.0019

6. Purbajanti, E.; Anwar, S.; Wydiati, F.K. Drought stress effect on morphology characters, water use efficiency, growth and yield of guinea and napier grasses. *Int. Res. J. Plant Sci.* 2012, *3*, 47. [Google Scholar]

7. Mwendia, S.; Yunusa, I.; Whalley, R.; Sindel, B.; Kenney, D.; Kariuki, I. Use of plant water relations to assess forage quality and growth for two cultivars of Napier grass (*Pennisetum purpureum*) subjected to different levels of soil water supply and temperature regimes. *Crop Pasture Sci.* 2014, *64*, 1008–1019. [Google Scholar]

8. Holm L, Pancho JV, Herberger JP, Plucknett DL, 1979. A Geographical Atlas of World Weeds. Toronto, Canada: John Wiley and Sons Inc

9. Charlotte Grossiord, Thomas N. Buckley, Lucas A. Cernusak, Kimberly A. Novick, Benjamin Poulter, Rolf T. W. Siegwolf, John S. Sperry, Nate G. McDowell

10. Irmak, S., and Mutiibwa, D. (2010), On the dynamics of canopy resistance: Generalized linear estimation and relationships with primary micrometeorological variables, *Water Resour. Res.*, 46, W08526, doi:10.1029/2009WR008484.